# Living through the Pandemic with a Disability: A Longitudinal Qualitative Study

Janice Chan [1], Somayyeh Mohammadi [1,2,3], Elham Esfandiari [1,2,4], Julia Schmidt [1,2], W. Ben Mortenson [1,2,5] and William C. Miller [1,2,5,*]

1 Department of Occupational Science and Occupational Therapy, Faculty of Medicine, University of British Columbia, Vancouver, BC V6T 1Z4, Canada
2 Rehabilitation Research Program, GF Strong Rehabilitation Centre, Vancouver, BC V5Z 2G9, Canada
3 Department of Psychology, Kingston University, London KT2 7LB, UK
4 Edwin S.H. Leong Healthy Aging Program, University of British Columbia, Vancouver, BC V6T 1Z3, Canada
5 International Collaboration on Repair Discoveries, Vancouver, BC V5Z 1M9, Canada
* Correspondence: bill.miller@ubc.ca; Tel.: +1-604-714-4108

**Abstract:** This study investigated the experiences of people with disabilities during the first year of the COVID-19 pandemic. Four semi-structured qualitative interviews were conducted individually with 13 participants between May 2020 and February 2021. The data were thematically analyzed. Three themes were identified: (1) "Being an active agent in changing how things are done in the face of COVID restrictions", revealed changes that participants made to their daily routines resulting from government-imposed and self-imposed restrictions; (2) "Pandemic restrictions wreak havoc", explained participants challenges with adapting to the restrictions; and (3) "Trying to be resilient in the face of pandemic changes" described participants' efforts to cope with life during the pandemic. The findings illustrate how life changed for people with disabilities during the pandemic. Participants reported specific types of challenges at each time point. As the vaccine rollout became more imminent, participants expressed more hope for the future and getting back to normal.

**Keywords:** individuals with disability; COVID-19; pandemic; health outbreak





## 1. Introduction

When the coronavirus 2019 (COVID-19) was declared a pandemic in March 2020, governments around the world imposed preventive measures to avoid the spread of the COVID-19 virus. In Canada, the government implemented public health guidelines detailing recommendations to curb the spread of the disease. Recommended practices included, but were not limited to, self-isolation, avoiding crowded areas, avoiding public and mass gatherings, and working from home. Research emerging post-pandemic shows that pandemic restrictions were associated with decreased physical and psychological well-being in the general population [1]. However, the physical and psychological impacts of the pandemic, such as restrictions in accessing medical facilities at the beginning of the pandemic and an increase in depression and loneliness, which have been found to result in poor outcomes [2], have been worse for marginalized and vulnerable populations [3], including six million Canadians who live with at least one physical disability [4]. People with disabilities face unique challenges during pandemics, disease outbreaks, natural disasters, and other traumatic events [5,6]. Because the COVID-19 pandemic was a new experience that no one in the world has ever experienced, our study aims to elucidate the specific, unique challenges that people with disabilities faced in this time period. So far, preliminary data suggests that during the COVID-19 pandemic, people with disabilities were at higher risk of infection and death than people without disabilities [7,8]. In addition, people with disabilities were less likely to access necessary medical support, such as getting appointments for general checkups, because healthcare systems prioritized their

resources for patients with COVID-19 [9]. Due to the shortage of medical supplies during this time [10], people with disabilities were unable to access things such as masks and gloves that are routinely required for persons without disabilities to provide personal care for persons with disabilities. Additionally, things such as ventilators were being given to COVID patients even if they were required for people with disabilities who did not have COVID. Furthermore, while some physical distancing guidelines can restrict the movements of both people with and without disabilities. However, the impact was higher for people with disabilities. For example, people who used mobility devices had much more difficulty using public transit because of changes to loading procedures made on a systemic level that resulted in a lack of assistance from drivers [11].

People with disabilities (PWD) had poor outcomes following the COVID-19 pandemic and its accompanying restrictions when compared to those of people without disabilities [2]. Therefore, it is crucial to investigate the impacts of the COVID-19 pandemic on the daily lives of people with disabilities to better inform the development of resources and guidelines to address their needs in future pandemics and outbreaks. The current knowledge regarding the impact of the COVID-19 pandemic is based on qualitative studies conducted during a restricted time frame of the pandemic [12]. However, the changes in pandemic restrictions and their impact on people with disabilities are an important factor that has not been studied. Hence, this longitudinal qualitative study explores how the COVID-19 pandemic impacted the daily lives of people with disabilities to understand the challenges that people with disabilities face and how they adjust their lives or show resilience in the face of the imposed restrictions over the first ten months of the pandemic as various health regulations were introduced.

## 2. Research Process

### 2.1. Participants and Methods

Data from this study were collected as part of a larger mixed-methods study. The protocol for this study has previously been published [13]. This study used a longitudinal research design and qualitative semi-structured interviews as the primary means of data collection. The research ethics board of the University of British Columbia approved the study. We used the consolidated criteria for reporting qualitative studies 2 (COREQ-2) to report the findings [14].

### 2.2. Eligibility and Recruitment

The study did not include or exclude on the basis of age, disability diagnosis, indigeneity status, or underrepresentation characteristics. The following inclusion criteria were used: Participants needed to (1) self-identify as having a mental or physical disability; (2) reside in BC during the study period; and (3) read and speak English. Advertisements were placed on social media sites such as Twitter and Facebook to recruit participants for this study. Participants from previous studies were also invited to take part. Organizations providing services to people with disabilities (e.g., Tetra Society of North America) shared the study advertisement on their social media sites and in their newsletters. Potential participants were asked to read the consent form and then review it with one of the study investigators. The investigator addressed any questions they had before completing the consent form online.

### 2.3. Data Collection

Four rounds of interviews were conducted between May 2020 and February 2021. The first round of interviews (T1) was conducted between May 2020 and June 2020; the second round of interviews (T2) was conducted between June 2020 and July 2020; the third round of interviews (T3) was conducted between August 2020 and September 2020; the final round of interviews (T4) was conducted in January and February 2021. The first three rounds of interviews were aligned with the first three phases of reopening and decreases in restrictions in British Columbia, Canada. The final round of interviews was conducted after the beginning of

vaccination for people with underlying health conditions in British Columbia. The co-authors and other members of the COVID research team developed the semi-structured interview guide based on Hammel's 2009 classification of occupations. Questions explored activities to connect and contribute, for restoration, and to connect the past to the future. The original semi-structured interview was modified three times to reflect the restrictions of the respective time points. Interviews were conducted via Zoom and lasted approximately 60 min. Participants were asked to answer questions related to the impact of the pandemic on their daily activities. While the main structure of the interview guide remained the same throughout the study, at the beginning of each data collection time point, the researchers updated and modified some questions to reflect on the new changes in the restrictions and the pandemic's progress. All interviews were audio-recorded using the record function in Zoom. The interviews were transcribed first using the Amazon Transcription service; transcribers then manually checked and edited each transcription for accuracy.

### 2.4. Data Analyses

Codebook thematic analysis was used to analyze the data, as it focused on identifying explicit and implicit meanings within the data. Codebook thematic analysis enable the researchers to develop themes early on. However, it also enables the researchers to review, refine, and develop new themes as data analysis progress [15].

To analyze the data, transcriptions were imported into NVivo (version 12) [16]. Before the coding, BM, CB, and WCM trained the coders, i.e., SM and JC. SM, JC, and BM coded the first three interviews (from the first round of interviews) separately and then met to compare the developed codes and resolve the discrepancies. After developing the first coding guide, SM and JC coded the rest of the interviews separately. After coding every two interviews, they conducted regular meetings to compare the codes, resolve the discrepancies, and update the coding guide. The updated coding guides were continuously shared with the research team to receive and integrate their feedback. The round-robin format was used, meaning that the last 40% of the interviews (of T1) were divided between the two coders. Then, each interview was mainly coded by one of the coders. The coders then sent the interviews that they coded to the other coder to review and amend if needed. The round-robin format was used for all the T2, T3, and T4 interviews.

The final version of the coding guide from T1 was used to code the T2 interviews. Then the coding guide was updated and modified to reflect the new codes based on the T2 interviews. The same procedure was used to code the T3 and T4 interviews, using the final version of the coding guide from the previous time point to code the interviews of the next time point, updating and amending the coding guide based on the new codes. Therefore, the final coding guide for each time point contained the codes from the previous time points and the new codes developed at that timepoint. In addition, coding guides reflected the number of times a specific code was used for each time point and whether there was an increase or decrease in using that code compared to other time points.

All the codes developed in T1 were posted on an online platform (miro.com) to develop the themes. Based on the content of each code and its description, the research team organized the codes into thematic categories. The content of each category was reviewed again to confirm its accuracy and relevance. Then the categories were organized into themes. After completing the codebook for each time point, if needed, the research team refined and updated the themes developed at the previous time point.

### 2.5. Trustworthiness Strategies

The study used two main trustworthiness strategies: triangulation and reflexivity. In terms of investigator triangulation, six different researchers with different backgrounds and levels of experience were involved in data collection and analysis of the interviews. Additionally, the initial codebook was developed in consultation with the senior researchers on the team and based on Hammell's classification of occupations [17]. More information about the interview guide can be found in [18]. Participant triangulation involved recruiting

participants with disabilities due to several conditions, such as cerebral palsy, amputation, and stroke. To facilitate reflexivity, interviewers recorded self-reflections following each interview. In terms of positioning, none of the authors or interviewers self-identified as individuals with disabilities. SM, JC, CB, JS, and TJ identified themselves as female, and BM and WCM identified themselves as male. None of the authors had a personal relationship with the participants. The rest of the interviews were conducted by a master's student in occupational therapy. All members of the research team were involved in rehabilitation research, which considers participants within their specific contexts. The research team's understanding is that people cannot be understood apart from their context, and vice versa. To build rapport between participants and the interviewers, the same participant was interviewed by the same researcher throughout the study. Participants were informed about the interviewers' education and occupation. Other personal information was not disclosed to the participants.

## 3. Results

### 3.1. Demographics

Thirteen individuals participated in the study. Eight participants reported their sex as female, six reported their gender as woman, five reported their gender as man, and two indicated they were non-binary. The average age of participants was 55.76 (SD = 16.72). Most participants ($n = 7$) had some college or university education; four attended graduate school, one graduated from college or university, and one participant attended high school. Only one participant reported mental disability, and the rest reported physical disability ($n = 12$). Regarding living situations, nine participants lived alone, and only four participants lived with a family member. The number of participants who were born in Canada was 11. Most participants in our study were either retired ($n = 4$) or received disability assistance ($n = 4$). The remaining participants were students p($n = 1$), self-employed ($n = 2$), or had other types of employment ($n = 2$). Pseudonyms have been used in this section to preserve the confidentiality of participants.

### 3.2. Final Codes and Themes and Their Development

Overall, the final codebook that was developed after T4 contained 23 subthemes. The subthemes were organized into three major themes evident across the four time points. Theme 1: "changing how things are done in the face of COVID-19 restrictions", explained various changes that participants had made to their daily routines resulting from government-imposed and self-imposed restrictions. Theme 2: "struggling with reactions to the pandemic restrictions". Theme 3: "trying to be resilient in the face of pandemic changes". The following will discuss the themes found in further detail. The changes in subtheme use for each Theme from T1 to T4 are shown in Table 1.

**Table 1.** The changes in using codes in each Theme from T1 to T4.

| Theme | Sub Theme | T1 | T2 | T3 | T4 |
|---|---|---|---|---|---|
| Being an active agent in changing how things are done in the face of COVID restrictions | Making adaptations in day-to-day activities due to COVID-19 restrictions | 63 | 22 | 19 | 3 |
| | Changing habits to protect themselves and others from COVID-19 | 39 | 46 | 20 | 21 |
| | Using technologies to connect with more people | 41 | 17 | 7 | 11 |
| | Providing support for others while going through the pandemic | 27 | 17 | 8 | 11 |
| | Receiving support from the community | 16 | 6 | 7 | 2 |
| | Defining their social bubbles | NA | 16 | 16 | 5 |
| | Not experiencing major changes in life | 20 | 3 | 11 | 16 |
| | Finding ways to fill the time | 29 | 18 | 12 | 12 |
| | Trying to mentally and physically stay healthy | 39 | 14 | 24 | 18 |

**Table 1.** *Cont.*

| Theme | Sub Theme | T1 | T2 | T3 | T4 |
|---|---|---|---|---|---|
| Pandemic restrictions wreak havoc | Experiencing restrictions on where to go | 44 | 13 | 23 | 11 |
| | Feeling anxious and isolated during the lockdown | 18 | 9 | 11 | 9 |
| | Missing social connections and activities | 27 | 4 | 11 | 13 |
| | Experiencing limited in-person interactions with outside the immediate family | 27 | 12 | 1 | 2 |
| | Having interpersonal challenges with others | 22 | 12 | 24 | 8 |
| | Highlighting perceptions of social inequality | 23 | 4 | 17 | 14 |
| | Experiencing challenges with societal and institutional responses related to the pandemic | 40 | 18 | 13 | 11 |
| Trying to be resilient in the face of pandemic changes | Being less busy | 14 | 6 | 9 | 0 |
| | Experientially enjoying day-to-day activities | 15 | 11 | 23 | 5 |
| | Feeling grateful for personal circumstances | 40 | 14 | 8 | 12 |
| | Experiencing challenges during COVID-19 pandemic and developing resilience | 61 | 2 | 4 | 3 |
| | Being optimistic about resuming pre-COVID-19 life after the pandemic | 17 | 10 | 5 | 22 |
| | Empathizing with others and what they go through during the pandemic | 10 | 1 | 2 | 2 |
| | Positive perceptions about reopening | NA | 27 | 27 | 10 |

### 3.3. Being an Active Agent in Changing How Things Are Done in the Face of COVID Restrictions

Participants described changes they made to their habits and daily routines due to the government's restrictions through Provincial Health orders from T1 to T4. As noted in Table 1, making adaptations in daily routines due to restrictions was one of the most frequently used subthemes at T1; however, as participants approached T4, the frequency of using this code decreased. Participants also adapted to protect themselves and others. For example, at T4, when masks were not mandatory in public, Joseph, 51 years old, disclosed his preference in the following the daily routines he obtained during the pandemic to protect himself as a person with a disability who is more vulnerable to COVID-19. He mentioned the following about wearing a mask in public:

> *I prefer to put it on once when I leave the house and then take it off once when I come home again. Which means that I am one of those people seen in open outdoor spaces still wearing the mask.*

Participants recounted adaptations to their lives based on the restrictions. For example, Wendy, 31 years old, at T3 explained her attempts to see others in person when possible while still following guidelines: "My church did a 'staycation'. So, with physical distancing in bubbles, wearing masks, and everything. We did a scavenger hunt, went for a quote-in-quote, 'hike'".

Another change participants made was the new or increased use of technologies such as Zoom™ to connect with others. This new long-demanded change became possible for people with disabilities due to the restrictions forcing everyone to use online tools to communicate, and therefore, people with disabilities could also increase the use of online tools in their everyday lives. As described by Amy, 40 years old, who pointed out how they used technology to stay connected with the LGBTQ+ community at T1, "All the pride events were going to be in person, and it's only because we can't have large crowds gathering that we're doing it online".

The previously mentioned change becomes more important when compared with the fact that people with disabilities asked for online tools to participate in work and social events; however, because the general public did not have the same needs before the pandemic, the need for people with disabilities to work online and use online tools was ignored. The comparison of society's response to the general public's needs during the pandemic results

in comparing the societal responses to people with disabilities with the societal responses to able-bodied people during the pandemic at T1. Pauline, 35 years old, at T1, expressed her frustration at society's slow responses to people with disabilities by indicating, "People making all of these changes to society and space in order to protect their health when disabled people have been asking for changes and have been told that these changes aren't possible". However, to a lesser extent, participants empathized with others based on what they saw and heard of others' experiences during the pandemic. This included acts of violence due to sex, race, or enforcement of government-imposed regulations.

Empathizing with others was mainly observed in T1 and was rarely used in the remaining time points. This sub-theme explains the ability of people with disabilities to consider what others are going through while they were considered an at-risk group during the pandemic. Participants described ways they provided support for others during the pandemic, albeit they were not volunteering in their roles prior to the pandemic. Some participants described checking in and picking up groceries for neighbors and friends. For example, Mary, 69 years old, at T4 explained:

*My friend, I was friend with friends with a family, basically. He, the husband, died, was dying of cancer, so I was helping them out a lot. I go shopping for them. So when COVID hit, I continued doing that for them and keep giving her support and keeping him company and just hanging around, so I've continued to do that after he died.*

Others described how services such as grocery delivery provided support for them during this time, as Emma, 60 years old, at T1, recounted the support she received from others by saying, "There's some specific groups here on the [location] and some Facebook groups that are willing to help if you need it. Pick things up and shop for you". The number of times people brought up receiving support from others in the community did not increase substantially from T1 to T4. While receiving and providing support were eventually used less to code participants' interviews, a bigger reduction was observed in receiving support from others as time progressed.

Throughout the pandemic, the government consistently asked people to keep their social circles small. T2 interviews showed a new subtheme that described participants' attempts to define their social bubbles. This subtheme described participants' efforts in defining their social bubbles and keeping those in their social bubbles safe. A negative case of "using technologies to connect with more people" also emerged at T2 as participants described "Zoom fatigue" and "not wanting to connect with people online".

The subtheme "not perceiving major changes in life" emerged as a negative case in this theme and described how some participants did not identify changes in their lives due to pandemic restrictions. While this subtheme has not been used frequently, it remained stable from T1 to T4. To describe not experiencing major changes in life, Henry, 68 years old, at T1, said, "I took it as a bit of a joke because it looked like everybody was gonna have to do what I do, spend a lot of time at home and you can't do much and all the rest of it". This sentiment was evident across all time points.

*3.4. Pandemic Restrictions Wreak Havoc*

Another emerging theme was "Pandemic restrictions wreak havoc". As time passed and BC moved towards reopening, we observed a reduction in the use of the subthemes under this theme. The highest number of times that subthemes under this theme were used was at T1, T3, and T4. There was a reduction in using this subtheme at T2, which was during phase 2 of BC reopening when some restrictions were temporarily lifted.

One of the most frequent subthemes at T1 described participants limitations in going to hair salons, traveling, or meeting with doctors and health professionals at all timepoints. For example, George, 59 years old, at T3, disclosed his frustration about the restrictions and ongoing closure of the services by indicating, "I couldn't go up shopping for anything, I managed to make her a card, but that was it, but hadn't gotten her anything". While some of these restrictions affected all people in Canada, some, including visiting doctors and health care professionals, had serious implications for people with disabilities and speak

of an important need of this group that was not met immediately by societal reactions, especially at the beginning of the pandemic.

Furthermore, participants described various negative emotions (including anxiety and isolation) that they experienced during the pandemic. Due to the restrictions, such as those on visitors to hospitals, participants felt anxious and isolated during the lockdown. With regards to restrictions on hospital visitors, Mary at T4 expressed frustration about the people who are dying alone in the hospital by explaining, "They won't let anybody in. They'll only let one family member a day in, which is horrible for people, and I figure, call me cynical, but people are dying anyway . . . let them be with their family".

The pandemic restrictions led to a feeling of isolation and internal turmoil that caused anxiety as participants expressed the desire to be a part of significant events and milestones such as family picnics and birthdays but also wanted to stay safe. In their interviews, participants emphasized the importance of relationships and the ways in which they tried to maintain those relationships. Mary summed up her experience regarding social life during the pandemic, explaining, "maintain your social connections as best you can through the Internet . . . , you know, no hugging . . . " The restrictions in place due to the pandemic caused participants to struggle both with finding ways to connect with people outside their homes and with finding ways to live with people within their homes. Amy at T3 described her concerns about the well-being of other family members by saying, "You can really only worry about yourself at the end of the day . . . I can worry about my parents too". Even when participants left their homes, they experienced challenges with other people related to what they saw and what others said to them. Joseph at T3 described the experience of going out as "strange . . . it seems like everyone on public transit now are all afraid of each other . . . ".

Another subtheme commonly used in this theme was struggling with societal responses to the pandemic. Specifically, participants divulged stories of how the policies implemented during the pandemic highlighted the inequities within society and affected people disproportionately depending on age, disability status, and economic status. Amy at T3 shared her opinion on how the pandemic affected some families with children more than others and explained how society did not support families who did not have resources to provide care for their children on their own:

> *There's people who already have someone at home who can stay with the kid or people who can afford tutors, and then I've heard of a couple of families within my connections who, the one of the parents is gonna quit their job to homeschool the child.*

Other participants experienced accessibility barriers, which are defined as any barriers that prevent people with disabilities from fully and equally participating in society (The Accessible BC ACT). In our sample, there were both participants who expressed gratitude for the British Columbian response to COVID-19, as captured by the subtheme "happy with the BC response," and participants who were feeling frustrated with what was happening. Related to social inequality, Emma at T3 pointed out:

> *I don't know how you would determine the need of someone who is disabled or at high risk, medical conditions. They absolutely don't have a choice. That can't get out and get groceries. Those are the ones that should be prioritized for deliveries. And I'm just really surprised that wasn't.*

### 3.5. Trying to Be Resilient in the Face of the Pandemic

Participants were challenged both by self-imposed restrictions and government-imposed pandemic related restrictions. Participants described being less busy and less motivated to complete their tasks initially, whereas at T4, the related subtheme was not used at all. For example, Mary at T1 reported, "The biggest one is motivation. Motivation to do the things that I know I would really enjoy doing. But just getting started at anything that that is, that's probably the hardest thing still I'm having to do deal with".

While participants' initial responses indicated they are less busy and have less work to do, they explained vigorous attempts to fill the time by learning new skills and starting up old hobbies. For example, Wendyat at T1, in response to how she is filling her time reported:

*I'm doing new things about technology, go for my own work, so I think I am already familiar with Zoom and whatnot, but speaking more familiar and change tools for online teaching. I also have learned about redesigning business model.*

While the significant peak in learning new hobbies was during T1 and the usage of that subtheme eventually reduced, even at T4, some participants acknowledged new attempts to learn new hobbies and fill their time; for example, Joseph at T4 explained: "I've been painting more. I was doing collages that were, I thought, a progression of the way I like to do art".

Participants also described numerous efforts to stay healthy, both mentally and physically, by staying active and practicing spirituality. One strategy that participants used to foster resilience was to experientially enjoy day-to-day activities. For example, participants described enjoying family time and cooking. Participants also described enjoying what they were allowed to do, using activities to their advantage, and utilizing physical activities such as kayaking to connect with others. For example, John, 81 years old, at T2, pointed out:

*I guess going out and just moving around more, so that's pretty good for well-being, I guess. Getting more active again, so floorball helps, a little bit of tennis helps. I think the floorball thing helped a lot. That's being more active than just moving around.*

Joseph at T4 explained how he is trying to improve his mental well-being by taking new courses:

*Actually, I'm taking a mindfulness-based cognitive therapy course. So just this week, in the third week of the course, you do three 3 min meditations every day. So I start my day with a meditation and end my day with a meditation.*

Another subtheme that has been used the most was related to participants' gratitude for their circumstances at all time points. They attributed their ability to cope with the numerous changes to the many privileges that they had. For example, Wendy at T1 disclosed:

*I can't really say that; like you said, I do live alone on the mainland, so I imagine I would experience more loneliness the way that a lot of people are. I'm quite content to spend time alone, but this is a long, long time. So I think that would have been different. I also like the place that I live on the island. I can go outside and, on the farm, and lots of space and [there is] quiet road near where we live, so I've been going for a lot of walks and maybe even walking more than I normally do.*

Another sub-theme primarily observed during T1 and suddenly used less often after is related to facing challenges and developing resilience. For example, John at T1 explained, "I accept those as being disruptions, and I'm hopeful that COVID will go away sometime, and they won't be there. But there's nothing much I can do about them now".

Participants spoke of being optimistic that pre-COVID-19 life would resume throughout the four time points. Some participants spoke hopefully of plans such as travel and attending music festivals. Other participants spoke of developing a mindset that allowed them to live life daily to foster the hope they needed to get through the pandemic. "Being optimistic about resuming pre-COVID-19 life after the pandemic" was used more at T4. For example, Henry at T4 described, "Maybe it'll be like flu and you get a flu shot. And that's what you do. You get the latest viral shot with your flu shot, maybe, it'll all be good".

Finally, the subtheme "positive perceptions about reopening" emerged in the second round of interviews and was observed a few times afterwards. These reflections included thoughts about experiencing some level of normalcy and feeling content about restarting previous activities and meeting others in person. Some participants had reservations embedded within their reflections on reopening, though, as they did not feel comfortable going to public places. The reflections included redefining what was normal and getting used to the new normal.

## 4. Discussions

This study aimed to explore the impact of the COVID-19 pandemic on the daily lives of people with disabilities. Three identified themes in our study provide an overview of the journey that individuals with disabilities went through, the new imbalances in their lives, and their attempts to show resilience. One of the main observations was their attempts to engage with the pandemic-related changes and restrictions. This finding was in line with similar studies showing that when an individual is at higher risk, they are more likely to follow the health guidelines to prevent catching the virus [19]. One of the factors that helped people with disabilities follow health guidelines and protect themselves was the support they received from the community. Community services have pivotal benefits for people with disabilities. These services aim to support people in their daily activities. During the pandemic, many community services stayed active and tried to maintain their services [20], which was an important source of support for the participants in this study. Participants in this study needed support from their family, friends, and community to overcome the challenges they were facing during the pandemic; they expressed an empathic understanding of the hardship that other people experienced during the same period. Learned matching theory [21] suggests that people acquire empathy through associative learning. Empathy can be learned based on personal experiences [22,23]. Therefore, it is likely that people with disabilities could identify the challenges and difficulties experienced by others and be able to express empathic responses.

One of the main concerns of the participants in this study was the restrictions they faced in accessing the healthcare system. During a natural crisis or a disease outbreak, most healthcare systems need to focus on addressing the needs of people affected directly, e.g., patients with COVID. Therefore, the needs of people with disabilities during the crisis may be ignored [24]. This is in line with other studies showing that, while even people without disabilities reported challenges accessing healthcare, people with disabilities reported specific challenges related to their disabilities [25]. The healthcare system's lack of resources to attend to the needs of people with disabilities may be associated with perceived discrimination among this group [26]. In the long term, this will result in a reduction in healthcare use among people with disabilities and, subsequently, increased disability and emotional distress [27]. Similarly, some participants expressed their frustration with society because they felt their request to have access to some accommodations, such as working from home, was disregarded. However, when the majority of people needed to have access to these services, society provided them in a short time.

Furthermore, participants reported experiencing interpersonal and societal challenges. These challenges were experienced because non-disabled individuals did not follow pandemic-related rules, e.g., getting too close to a person with a disability in a grocery store. Participants in our study disclosed that, on some occasions, they were forced to remove themselves from the situation. Ignoring the needs of people with disabilities by non-disabled individuals can contribute to isolation, exclusion, and perceived discrimination among people with disabilities [27]. Therefore, educating people without disabilities on their responses to people with disabilities, especially during a crisis, is essential.

As we moved through the pandemic, participants expressed optimism about resuming life before the pandemic. The observed optimism had characteristics of "dispositional optimism", which is defined as optimism that good events will occur in the future [28] Dispositional optimism can prevent mental and physical health challenges after experiencing a disaster [28]. During the pandemic, the government's and healthcare providers' messages were constantly related to resuming normal life after the development and roll-out of the vaccine. Observing more optimistic quotes from the participants was in line with the breaking news related to developing a potential vaccine for COVID-19. This could indicate the impact of positive government messages on the well-being of society's members. However, a recent alternative theory explains that having high expectations about the future can harm one's well-being if those expectations are never met or achieved. Fletcher [29] suggested that individuals with higher levels of hopefulness may experience

lower well-being in circumstances of high stress. Therefore, while developing hope and optimism can be essential for society's overall well-being, it is important to be mindful of the potential negative impacts of developing an unrealistic level of hope.

This study had some limitations. The first interview happened during the first lockdown in British Columbia, and therefore, we do not have information on the participants' life experiences before the pandemic. Our sample was restricted to residents of British Columbia; this is an important consideration when transferring the findings to other jurisdictions. While this study had some limitations, it also had several strengths. By recruiting a sample of participants who self-identified as people with disabilities, the sample included a wide range of physical and psychological disabilities. Furthermore, the longitudinal nature of this study allowed us to follow up with participants for almost a year after the beginning of the COVID-19 pandemic. This provided an opportunity to observe the variations in participants' lives during the first year of the pandemic and how they adapted to the changes in their lives. In addition, our findings highlighted the importance of the perception of equality in the lives of individuals with disabilities and their request to have the same opportunities as others during and after the end of the pandemic. Finally, our study pointed out the critical aspects of optimism and hope. While this study cannot determine the optimal level of optimism and hope, policymakers and health care providers need to be careful about the nature of the messages they convey to the public, especially to people with disabilities.

## 5. Conclusions

This study is one of the few that provides a unique perspective on the lives of people with disabilities during the first year of the pandemic by applying a longitudinal qualitative design. People with disabilities were not exempt from the issues that people without disabilities faced during the first year of the pandemic; however, they experienced issues in addition to those experienced by people without disabilities. For example, people with disabilities had difficulty with public transit due to drivers not being able to assist with securing their wheelchairs [5,6,12]. The findings suggest that any new changes in the restrictions created an urgency to mobilize efforts to readjust lives with the new restrictions. However, similar to previous studies, participants in this study showed resilience when facing rapid changes caused by the pandemic [30]. In addition, hope and optimism were important factors in the well-being of these individuals. Based on the findings of this study, we recommend that accessible tools and community support developed during the pandemic remain in place so that people with disabilities can benefit from them. Furthermore, healthcare providers and governments must meet the needs of people with disabilities during future pandemics, disease outbreaks, and national disasters. More importantly, the government should educate society on how to interact with and engage with people with disabilities during a crisis. Finally, the government must find a balance between sending a message of hope and optimism and reality.

**Author Contributions:** Conceptualization: W.C.M., W.B.M., S.M. and J.C.; methodology, W.C.M., W.B.M., S.M. and J.C.; formal analysis, W.C.M., W.B.M., S.M., J.C., E.E. and J.S.; writing—original draft preparation, S.M. and J.C.; writing—review and editing, W.C.M., W.B.M., S.M., J.C., E.E. and J.S.; supervision, S.M., W.C.M. and W.B.M. All authors have read and agreed to the published version of the manuscript.

**Funding:** This research has not been funded.

**Institutional Review Board Statement:** The study was conducted in accordance with the declaration of Helsinki and approved by the ethics committee of the University of British Columbia (H20-01109).

**Informed Consent Statement:** All participants read and signed the informed consent prior to their participation.

**Data Availability Statement:** Due to the qualitative nature of this study, participants only agreed to share some of their quotes in research papers. Therefore, the complete interviews are not available.

**Acknowledgments:** We would like to thank all our participants and 41 team members, including undergraduate and graduate students at the University of British Columbia, who devoted their time to this project (which is part of a bigger project) during the first two years of the pandemic.

**Conflicts of Interest:** The authors declare no conflict of interest.

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
