# Peer review of "Living through the Pandemic with a Disability: A Longitudinal Qualitative Study"

_disabilities, doi:10.3390/disabilities3030020_

Round 1

Reviewer 1 Report

The manuscript needs significant revisions.

Author Response

Comment 1. As an overall comment on the introduction, you stated that it was “…difficult to understand the knowledge gap , particularly in the Canadian context,  to establish the urgency of the issue and thus, the criticality of the study.”

Response: Thank you for your comments. While there is a lot of research that has looked at how the COVID-19 pandemic and its accompanying restrictions impacted people who did not identify as having a disability, this study looks specifically at the experience of PWD’s. It is critical to study this immediately following this pandemic as the findings can help to better inform government policies that would support PWD’s if we were to ever be in a pandemic again.

Comment 2. Line 29: A bit confusing to read, coronavirus disease 2019 pandemic. Please revise

Response: This line (page 1, line 26) has been revised to read “at the beginning of the coronavirus 19 (COVID-19) pandemic in March 2020…”

Comment 3.  Line 31-33: When we say government instructed to follow, are we saying that government introduced certain public health measures and forced people to use them? If social distancing guideline is an instructive measure, then how come mass gathering is a recommended measure. Having clarity of public health measures, what was their purpose, and what kind of measures were introduced will be helpful.

Response: The difference between an instructive measure and a recommended measure was clarified in the new draft of the paper. Please see page 1, lines 28-31. The lines now read “... the government implemented public health guidelines detailing recommendations to curb the spread of the disease.”   

Comment 4. Line 34: What do we mean by emerging research. Clarity if terminology is needed as we are now living in post-pandemic times.

Response: “Emerging research” was revised to read “research emerging post pandemic” to acknowledge that we are now living in post-pandemic times (page 1, line 31).

Comment 5: Line 43: Why they are less likely to access the supports? Ain't medical system already overburdened?

Response: Overwhelmed has been replaced with “prioritize” in the manuscript (page 1, line 40).

Comment 6: Line 45-46: Confusing to read, 'younger and healthier individuals' being triaged

Response: This sentence (page 1, line 42) was revised to read “some healthcare systems allocate their resources to address the needs of younger and healthier individuals over those of people with disabilities.”

Comment 7: Line 65-67: Do you mean longitudinal research design using qualitative semi-structured interviews?

Response: This was clarified in the new draft. The lines (page 2, lines 69-72) now read “This study used a longitudinal research design and used qualitative semi-structured interviews as the primary means of data collection. The data were analyzed using qualitative analyses.”

Comment 8: Line 72: Please describe how many self-identified mental and physical disabilities

Response: Only 1 participant reported mental disability. This information has been added to the results section of the paper (page 4, line 164).

Comment 9: Participant inclusion criteria is unclear. Did the study consider, age, disability diagnosis range, Indigeneity status, under-representation characteristics, gender etc.

Response: Participant inclusion criteria is described in the Eligibility and Recruitment section of the paper. Age, disability, diagnosis range, Indigeneity status, under-representation characteristics, and gender were not considered as part of the inclusion/exclusion criteria. This is also clarified in the eligibility and recruitment section of the paper. Please see page 2, lines 75-78.

Comment 10: Were accommodation made to include the needs and priorities of participants before and during the interviews?

Response: Interviews were conducted in the home and all interviewers ensured participants felt safe and were comfortable for the duration of the interview. Because the interviews occurred in the home, no specific accommodations were provided for the participants.

Comment 11: Author must inform how the interview guide was developed and established.

Response: The following has been added to the manuscript to address the development of the interview guide (page 2, lines 95-99): “The co-authors, and other members of the COVID research team developed the semi-structured interview guide based on Hammel’s 2009 classification of occupations. Questions explored activities to connect and contribute, for restoration, and to connect the past to the future. The original semi-structured interview was modified three times to reflect the restrictions of the respective time points.”

Comment 12: Line 90: What was the average length of the interviews?

Response: Interviews lasted on average 60 minutes. This information was added to the manuscript. Please see page 3, Line 100.

Comment 13: Line 99: The author here says thematic analysis was used to analyze the data and in the above section, longitudinal qualitative method. This sounds confusing.

Response: Thank you for your comments, we clarified the above line (page 2, lines 69-72). The lines now read, “This study used a longitudinal research design and used qualitative semi-structured interviews as the primary means of data collection. The data were analyzed using qualitative analyses.”

Comment 14: Line 133: How does involvement of researchers performing the data collection lead to trustworthiness? Was there any one else who overlooked or consulted for the initial codes and codebook, the early analysis and key themes.

Response: Six different researchers with different backgrounds and levels of experience conducted the interviews. Additionally, the initial codebook was developed in consultation with senior researchers on the team. Please see page 3, lines 141-145.

Comment 15: Overall impression on results: The results must give the unique perspectives of PWDs based on the type of disabilities they had. The current results seem a bit generalized and less applicable to PWDs.

Response: The interviews focused on understanding the experiences of people with disability during the pandemic. The questions did not ask the participants to discuss how their disability influenced their experience but mainly focused on their experiences in general. Therefore, while some quotes discuss unique challenges as a person with a disability, this was not always the case. We revised our result sections and added more discussions to the sub-themes linked to their experiences as people with disability living through the pandemic. We also modified the themes and sub-themes.

Comment 16: Line 175: It is not recommended to use the names of participants and their ages so to preserve their confidentiality.

Response: Thank you for your comments. All the names mentioned in the manuscript are pseudonyms. We clarified this on (page 4, line 169).

Comment 17: Line 178 - 180: Why did they wear the mask, what was unique about it?

Response: This has been addressed in the manuscript. Please see page 5, lines 188 - 191. PWD’s are more vulnerable to COVID-19.

Comment 18: Line 186: Almost all of us went online to conduct meetings. What was unique for persons with disabilities?

Response: It is correct that people going online to conduct meetings was a societal response however, for persons with disabilities, this experience allowed them to expand the number and quality of connections they had with other persons with disabilities. We added an explanation on page 6, lines 201 - 212.

Comment 19: Line 199 - 205: Irrelevant finding given the scope of the paper.

Response: These lines have been removed per your suggestion.

Comment 20: Line 206- 215: Is it relevant to the talk about this experience of pwds during the pandemic and given the objective of the manuscript?

Response: This specific experience details the ability of persons with disability to consider what others are going through while they themselves are considered to be people at risk. Therefore, we believe this is an important topic to discuss, especially knowing that most research focuses on how able-bodied individuals show empathy towards people with disability and not the other way. We are hoping that this piece of information will further open up the discussion on investigating empathy in people with disability. We tried to be more clear about the importance of this topic on page 6, lines 222 – 228.

Comment 21: Line 237: Theme title is hard to understand and interpret. Please change it to something simpler so it follows the results in this theme.

Response: The theme title has been changed to “pandemic restrictions wreak havoc.”

Comment 22: Line 270-280: There is disconnect between the code and the further explanation and the quote. The point of pandemic policies and inequities for PWD is not articulated effectively,

Response: The descriptions of the codes have been modified to explain this better.

Comment 23: Theme 3.5: Above you spoke about how participants adjusted to the challenges brought about by the pandemic and policies, and this theme described what they did in general to remain resilient in life. Please revise it as per the objective of the manuscript and the population group's unique challenges.

Response: The objective of the manuscript has been modified to include the development of resilience. In addition, two theme titles have been modified to reflect the relationship between the development of resilience and the pandemic.

Comment 24: Overall impression on discussion:

  1. The discussion is not tied to the study objective and does not completely follow the results.
  2. The discussion could be tied to the existing literature more effectively.
  3. Discussion could focus on the things that governments could do

Response: We tried to make the discussion more concise and focused on the objectives of the study. We also removed sections that were not directly in line with the objectives of the study. We added more recent literature to the discussion and suggestions for the government. Please see page 9, lines 385 – 396; page 10, lines 397 – 421; page 11, lines 462 – 468.

Reviewer 2 Report

The manuscript I have been asked to review presents a four-wave longitudinal qualitative study focused on the experience of people with disabilities (PWDs) during the Covid-19 pandemic.  I am a personality psychologist and, like the authors, have studied the experiences of PWDs over the course of the Covid-19 pandemic using both qualitative and quantitative methods.  As a result, I was delighted to review the present manuscript.

Without a doubt, the focus of this study is worthwhile and timely.  As the authors note, PWDs have faces increased discrimination, as well as authentically existential threats in the face of the upheavals caused by the pandemic.  I applaud the longitudinal study design and the careful attention to participant-interviewer pairing over the course of this study.

I have several recommendations that I believe would substantially strengthen the current manuscript.

First, and most importantly, I found the data analysis in this study overly summative and would have liked to see the authors push their work towards more interpretive insights.  The three primary thematic clusters identified (changed in how things are done in the face of Covid restrictions, struggling with reactions to the pandemic restrictions, and trying to be resilient in the face of pandemic changes) seem universal and straightforward – if someone didn’t know anything about the experiences of PWDs during the pandemic, they might guess these three categories.  From my perspective, this kind of summative work is the necessary first step in qualitative analysis, but it is rarely sufficient to be the final step.  I did not feel that I had learned anything specific or substantial about the participants in this study.  Indeed, while the authors do an excellent job in the introduction and conclusion to describe the specific challenges (and strengths!) of PWDs in the face of the pandemic, the results section did not feel particular to PWDs.  For example, “making adaptations in daily routines due to restrictions,” called out as one of the most frequently used codes at T1 (line 172) was universally true.  Everyone had to make adaptations to daily routines in May-June 2020, so I did not feel that this insight helped me understand anything important about PWDs.  Without a doubt, commonalities between able-bodied people and PWDs during the pandemic are worth highlighting, but my own read of the broader literature (and my own qualitative work) suggests that PWDs faced distinctive challenges and brought distinctive assets to this experience and I did not feel that I got a window into this specificity in the manuscript.  In the Discussion, the authors briefly offer an example about perceived ableist actions (line 392), but this was an important exception to the overall presentation of results.

The authors importantly note their efforts to bring a reflective approach to the scholarship.  Reflexivity is an essential component of qualitative research.  I did not feel that the authors’ discussion of their own positionality helped me understand how they applied reflexivity in this study.  The authors note their gender identity and their identification as able-bodied people, but these statements of demographic characteristics do not help the reader understand the distinctive perspectives of these authors interpreting these data.  I believe positionality statements must be more than summative, but offer interpretive insights about the research encounter.

Finally, I would like the authors to explain how the thirteen participants included in this study were selected from the broader mixed-methods study (line 64).  Why were these the participants under investigation here?

In sum, I found this study to be an excellent start focused on an important topic.  I would like to see the authors push deeper and more specific in their analytic work.

Author Response

Comment 1. The results did not seem specific to persons with disabilities.   

Response: The focus of the interviews was on understanding the experiences of these groups during the pandemic. The questions did not ask the participants to discuss how their disability influenced their experience specifically. Where applicable, we tried to add quotes that focus on disability. Furthermore, some sub-themes were directly related to disability, such as "highlighting perceptions of inequality” and “experiencing challenges with societal and institutional responses related to the pandemic.” We tried to modify the wording throughout the result section to highlight those sub-themes focused on participants’ experiences as a person with a disability.

Comment 2. Why were the 13 participants in this study chosen from the participants of the larger study?

Response: Thank you for your comment. In the larger study, we recruited participants with spinal cord injury, traumatic brain injury, older adults, people with a disability (excluding SCI and TBI), and families with young children. The current manuscript only focused on the participants in the disability group. This group's interviewers, coders, and main team members differed from other groups. However, the semi-structured interviews were similar for all the groups.

Reviewer 3 Report

Thank you for your qualitative work. 

This is information that needs to be documented and shared with a wider audience. Your Discussion section was particularly informative and well balanced. 

Two points of consideration:

- Please indicate the type of qualitative research design that you used, such as phenomenology. I understand you did a thematic analysis as your methodology. 

- There are several typos in Table 1 under Trying to be resilient theme.

Author Response

Comment 1: Please indicate the qualitative research design used in the study

Response: This has been indicated in the methods section of the manuscript. Please see page 2, lines 69-72.

Comment 2: Revise typos in Table 1

Response: Thank you for your comment. We edited the typos.

Reviewer 4 Report

The intorduction section need further develpment, many research has been done so far about COCID 19 and I would like to see more references in this area. Also the article should state the originality and the added value of this research.

Author Response

Comment 1: The introduction needs further development

Response 1: We tried to modify the introduction based on the comments we received from other reviewers.

Comment 2: State the originality and value of the manuscript

Response 2: We modified the conclusion section to emphasize the value of the current manuscript. Please see page 11, lines 456 and 462 – 469.

Round 2

Reviewer 1 Report

Attaching the file for author's reference. 

Author Response

Please see attachment, thank you. 

Reviewer 2 Report

I did not observe substantive responses to my concerns on the initial submission.  I still find the results overly summative and not interpretive.  I still find the positionality statement to be too role-based and not perspective-based.

Author Response

Please see attachment, thank you. 
